# Single Radiotherapy Fraction with Local Anti-CD40 Therapy Generates Effective Abscopal Responses in Mouse Models of Cervical Cancer

**DOI:** 10.3390/cancers12041026

**Published:** 2020-04-22

**Authors:** Jana Wood, Sayeda Yasmin-Karim, Romy Mueller, Akila N. Viswanathan, Wilfred Ngwa

**Affiliations:** 1Department of Radiation Oncology, Dana-Farber Cancer Institute, Brigham and Women’s Hospital, Boston, MA 02115, USAwngwa@bwh.harvard.edu (W.N.); 2Department of Immunology, University of Veterinary Medicine and Pharmacy in Kosice, 04181 Kosice, Slovakia; 3Department of Radiation Oncology, Harvard Medical School, Boston, MA 02115, USA; 4Data Analysis and Modeling in Medicine, Mannheim Institute for Intelligent Systems in Medicine (MIISM), Heidelberg University, 69117 Heidelberg, Germany; 5Department of Radiation Oncology and Molecular Radiation Sciences, Johns Hopkins Medicine, Baltimore, MD 21287, USA

**Keywords:** cervical cancer, abscopal effect, in situ vaccination, radiotherapy, anti-CD40, immunotherapy

## Abstract

Current treatment options for advanced cervical cancer are limited, especially for patients in poor-resource settings, with a 17% 5-year overall survival rate. Here, we report results in animal models of advanced cervical cancer, showing that anti-CD40 therapy can effectively boost the abscopal effect, whereby radiotherapy of a tumor at one site can engender therapeutically significant responses in tumors at distant untreated sites. In this study, two subcutaneous cervical cancer tumors representing one primary and one metastatic tumor were generated in each animal. Only the primary tumor was treated and the responses of both tumors were monitored. The study was repeated as a function of different treatment parameters, including radiotherapy dose and dosing schedule of immunoadjuvant anti-CD40. The results consistently suggest that one fraction dose of radiotherapy with a single dose of agonistic anti-CD40 can generate highly effective abscopal responses, with a significant increase in animal survival (*p* = 0.0004). Overall, 60% of the mice treated with this combination showed long term survival with complete tumor regression, where tumors of mice in other cohorts continued to grow. Moreover, re-challenged responders to the treatment developed vitiligo, suggesting developed immune memory for this cancer. The findings offer a potential new therapy approach, which could be further investigated and developed for the treatment of advanced cervical cancer, with major potential impact, especially in resource-poor settings.

## 1. Introduction

According to the World Health Organization (WHO), cervical cancer is the fourth most common cancer in women worldwide, with almost 90% of deaths occurring in low- and middle-income countries (LMICs) [1,2]. In LMICs like India and in sub-Saharan Africa, cervical cancer is the second leading cause of cancer among women and the leading cause of female cancer deaths in Africa. Despite a lower incidence in high income countries (HICs), disparities in cervical cancer incidence and mortality are also pervasive in the United States [3], strongly influenced by factors such as race and socioeconomic status [4].

Radiotherapy presents a major treatment modality for cervical cancer, with treatment prescribed for both curative and palliative care to ease pain connected with tumor progression. In addition, radiotherapy may present the only curative option for patients with inoperable tumors [5]. In practice, patients with stage IIB-IVA cervical cancer are treated with a combination of external beam radiotherapy, brachytherapy and chemotherapy [6]. Combined radiotherapy regimens offer maximal local control of cancerous lesions, while sparing the healthy tissues and extending the survival of patients [1,7]. However, for most women in LMIC and other resource-poor settings who are disproportionately affected and diagnosed at an advanced stage, socio-economic barriers influence patient compliance with many fractions of radiotherapy and/or chemotherapy [8,9]. More accessible therapeutic approaches are needed for many patients with advanced disease.

One possible approach, investigated based on Mole’s observation in 1953, is that radiation can induce tumor shrinkage, not only in primary irradiated tumors, but also in distant metastatic tumors beyond the radiation field [10]. This rare phenomenon, known as the Abscopal effect, has been linked to mechanisms involving the immune system, which mediates the regression of tumors throughout the body [11]. The activation of the immune system is induced by the radiation damage of cancer cells leading to specific type of apoptosis, immunogenic cell death [6]. This leads to an increased release of tumor antigens and damage-associated molecular patterns, that in the presence of antigen presenting cells (APCs), could prime the abscopal effect [12]. Today, the growing consensus is that combining radiotherapy with immunoadjuvants could boost the abscopal response or cure rates for different cancers.

One rational choice for an immunoadjuvant to boost the abscopal effect is anti-CD40 therapy. CD40 is costimulatory protein binding to CD154 (CD40L) on T_H_ cells, which lead to the activation of APCs and a cascade of immune effects. Studies have reported on the overexpression of CD40 in human papilloma virus-infected lesions and advanced squamous carcinomas of the cervix, compared to very low levels in normal cervical epithelium [13,14]. CD40 ligation has resulted in a tumor growth-inhibitory effect, delivering potent apoptotic signals to carcinoma cells. This provides a potential therapeutic target that could generate antigens for boosting the abscopal effect [13]. Second, independent of the need for expression of CD40 on cancer cells, agonistic anti-CD40 can be potent stimulants of the immune system, with significant potential synergy in combination with other treatments [14]. This is because of the ability of agonist CD40 to activate antigen presenting cells (APCs), for the priming of antigen specific T cells and its capacity to redirect tumor-infiltrating myeloid cells with anti-tumor and anti-fibrotic activity. Recent studies have shown that radiotherapy can generate neoantigens that broaden the oligoclonality of the T cell response, presumably with the APCs helping to induce T cell responses against a wider array of tumor antigens [15]. This justifies a hypothesis that rational use of radiotherapy with the immunoadjuvant anti-CD40 can significantly boost the abscopal response or cure rates for advanced cervical cancer.

As a first step to investigate this hypothesis, this study investigates whether one fraction of radiotherapy combined with anti-CD40 can boost the abscopal effect in mouse models of advanced cervical cancer. The investigations involve titration of radiation doses in combination with anti-CD40 and different dosing schedules of anti-CD40. We investigate the in-situ administration of the anti-CD40 given well-known systemic toxicity limitations, and the minimal invasiveness of such a treatment approach for cervical cancer. The clinical translation of such an approach could potentially offer a more readily accessible treatment option for patients in low-resource settings, by implementation with brachytherapy, as we have reported in recent work [9].

## 2. Results

### 2.1. A Single Fraction of Radiotherapy Dose in Combination with In Situ Anti-CD40 Generates the Abscopal Effect, Inhibiting Tumor Growth in Both Irradiated and Unirradiated Tumors

Figure 1 shows the timeline, radiotherapy set-up, and representative corresponding dose volume histogram developed in treatment planning. Each animal had two tumors representing the primary and metastatic tumor. For all radiotherapy experiments, the SARRP (Small Animal Radiation Research Platform) was used to deliver one fraction of image-guided radiotherapy dose of 6 Gy to one tumor, while minimizing the dose to healthy tissues, with no dose to the contralateral tumor representing metastasis.

To assess the abscopal effect, mice with two tumors were randomized into four different cohorts, including: control group, group treated with radiotherapy only, group treated with anti-CD40 only and the fourth group, treated with one fraction of radiotherapy and the immunoadjuvant anti-CD40. In the first study, tumor volume measurements were carried out over two to three weeks to monitor the treatment response.

Figure 2 highlights the results demonstrating the abscopal effect, where only the group treated with radiotherapy and anti-CD40 showed significant therapeutic response in both treated and untreated tumors compared to the control cohorts. A similar observation of response in the combination group was made in other studies including those in studies below varying different parameters. The preliminary immune cell population analysis (Appendix A) also supports these observations. Interestingly, the use of anti-CD40 alone showed inhibition in tumor growth in the untreated tumor, but the primary tumor grew progressively and mice reached their end point in tumor size, following the IACUC approved protocol. The observed response in the untreated tumor has also been observed for other cancers with anti-CD40 administered in situ [16,17] and may be attributed to levels of immunosuppression in primary and metastatic tumor environments.

### 2.2. Radiotherapy Dose of 6 Gy Shows an Equal or Better Abscopal Response than Doses of 10 Gy and 15 Gy

In a separate study, the abscopal effect was investigated as a function of the radiotherapy dose: 6 Gy, 10 Gy and 15 Gy. As shown in Figure 3a, the treated tumors with various doses of RT boosted with anti-CD40 showed significant local responses in the treated tumors, compared to control mice for all doses. However, only the lowest RT dose of 6 Gy boosted with a single dose of immunoadjuvant induced a significant abscopal response in non-treated tumors (Figure 3b). The abscopal effect was demonstrated by delayed tumor growth on both sides and even complete tumor regression in 40% of the mice in the 6 Gy fraction cohort. The mice in the control group developed tumor ulcerations, which met the IACUC humane endpoint.

The observation of the abscopal effect when using a lower dose of radiotherapy has also been observed in previous studies with pancreatic cancer [16]. A potential explanation proposed for these observations is that high doses of radiotherapy can be immunosuppressive, including the potential for the increased death of peripheral APCs crucial for priming the abscopal effect. Ongoing studies are investigating this further.

### 2.3. Significant Increase in Survival Observed When Combining Radiotherapy with Anti-CD40 in Mouse Models of Advanced Cervical Cancer

We also investigated different dosing schedules for anti-CD40, administered either as a single dose or in three equal doses in a weekly interval. Figure 4a shows that the groups where RT (6 Gy) was included had a positive result on the treated tumor growth, but without a large difference among the various dosing schedules. Figure 4b indicates that the higher initial dose of anti-CD40 (here as a single dose of 20 µg/tumor/mouse) immediately post-irradiation can induce an abscopal immune response, with significant tumor size reduction on both sides. The abscopal effect was observed with complete tumor regression in 60% of mice in the group treated with RT and a single dose of anti-CD40. Complete tumor elimination was not observed in any other group. This finding was consistent with a significant increase in overall survival, comparing 21 days in control group to over 100 days in the combined group (*p* < 0.0004), as illustrated in Figure 4c, with the good health status of the mice reflected as weight gain (Figure 4d). The CT images (Figure 4e) two weeks post treatment further illustrate these findings. After 100 days post treatment, completely cured mice were subcutaneously re-challenged with 1 × 10^5^ TC-1 cells. After a few days, vitiligo was visible on the injected side (Figure 4f), but no tumor growth compared to the control, suggesting the development of tumor specific immune memory.

## 3. Discussion

Overall, the results of repeated studies consistently showed that one dose of radiotherapy in combination with the immunoadjuvant anti-CD40 can generate an effective abscopal effect, causing significant tumor growth inhibition or complete regression, in both the locally treated and untreated tumors. A consensus from recent studies suggests the potential mechanism of the abscopal effect illustrated in Figure 5.

The radiotherapy results in the release of neoantigens and damage-associated molecular patterns (DAMPs) lead to an increased infiltration of immune cells, mainly antigen presenting cells (APCs) [17]. Smilowitz et al. have reported better treatment outcomes with higher RT doses (15–22.5 Gy), with an adjunct of immunotherapy in advanced intracerebral melanoma [18]. Studies by the Formenti group have suggested that high doses of radiotherapy beyond 15 Gy lead to sub-optimal results in breast cancer [19]. Meanwhile, abscopal responses were also induced at lower single RT doses, e.g., in pancreatic cancer (5 Gy) [17] or lung cancer (6 Gy) [20]. As in previous studies, lower doses may be favorable to reduce immune-suppression. Therefore, a higher radiation dose may be counterproductive in regards to the activation of the abscopal mechanism on distant metastatic tumors. The APCs present in the tumor microenvironment might be killed by high RT, hence the antigen presentation could be stifled. On the other hand, lower RT doses generate less DAMPs, but also retain more APCs that could induce a more robust immune response, leading to the elimination of tumor lesions. Apparently, different types of cancer may require a different range of RT doses to induce optimal immune stimulation. More studies investigating immune-cell populations and other mechanistic studies should provide more insights. Studies varying the immunoadjuvant dose could also further help optimize the results obtained in this study.

In this study, consistently effective abscopal responses were observed in treatment groups, where anti-CD40 was administered at a higher single dose immediately after irradiation. It is possible that other immunoadjuvants could also engender optimal responses. FDA approved immune checkpoint inhibitors (ICI), blocking PD-1, PD-L1/PD-L2 or CTLA-4, have shown strong results in melanoma and Hodgkin lymphoma, but the gynecological cancers, including cervical cancer, have had significantly fewer responders to immunotherapy-only treatments; concurrent conventional fractionated radiotherapy with immunotherapy trials are currently accruing [21]. To date, there is no evidence of an abscopal response in an advanced or recurrent cervical cancer due to the combination of RT and ICI. However, a rational combination of RT with these ICIs may be beneficial.

Disparities in cervical cancer incidence and mortality have become increasingly apparent between HICs and LMICs, with high incidence and mortality prevalent in resource-limited settings, with limited access to treatment. The results in this study offer a potential treatment option that could be further investigated to improve the outcome, using one fraction dose of radiotherapy and immunoadjuvant. Access to radiotherapy is a major problem in LMICs and recent studies have shown that a reduced number of treatment fractions could significantly expand access to radiotherapy treatment for breast and prostate cancer [22]. One could anticipate similar benefits for cervical cancer in using single or hypofractionated radiotherapy. Furthermore, given the high upfront costs for the establishment of external beam radiotherapy, Viswanathan and co-authors have highlighted the potential benefit of using brachytherapy alone in cervical cancer treatment [9]. Such an implementation could be investigated with anti-CD40 as an immunoadjuvant, thereby extending the use of radiotherapy to treatment of both local and advanced/metastatic cancer, with curative intent leveraging the abscopal effect. The results in this study justify further investigations.

## 4. Materials and Methods

### 4.1. Cell Culture

The mouse cervical cancer cell line TC-1, expressing HPV16 E6/E7 with C57BL/6 mouse background, was kindly provided by Dr. T.C. Wu (Johns Hopkins Medical Institutions, Baltimore, MD, USA). The cells were cultured in RPMI-1640 media supplemented with 10% FBS, 1% penicillin/streptomycin (10,000 U/mL penicillin; 10,000 µg/mL streptomycin, respectively), 1% sodium pyruvate, 1% non-essential amino acids, 10 mmol/L HEPES and were grown in a humified incubator at 37 °C under 5% CO_2_ atmosphere.

### 4.2. Animal Studies

Naïve healthy 8- to 10-week-old female C57BL/6N mice were purchased from Taconic Biosciences (Hudson, NY, USA) and were housed in the Dana-Farber Cancer Institute’s animal facility. Mice were held in groups of five in standard cages under a 12 h light/dark cycle, with access to food and water ad libitum. All animal procedures were conducted according to the protocols approved by Institutional Animal Care and Use Committee (IACUC) at Dana-Farber Cancer Institute (protocol 15-040 approved on 5 January 2016), in compliance to the guidelines and regulations for the proper use and care of laboratory rodents. To mimic metastatic cervical cancer models, TC-1 cells were subcutaneously injected with 50 µL of PBS cells suspension at concentrations of 1 × 10^5^ cells/tumor, into both dorso-lateral flanks of each mouse. Mice were regularly monitored for tumor growth until the tumors reached the treatable size of 3–4 mm in diameter. The mice were randomized and divided into groups according to the study designs below. In all studies, tumor measurements and body weight or mice survival were monitored at least twice a week, to observe the efficacy of the treatments.

### 4.3. RT Dose Titration Study

Mice were randomized into four groups (5 mice/group): control and 3 groups with different a single radiation dose (6 Gy, 10 Gy and 15 Gy). The irradiated mice were additionally intratumorally injected with anti-CD40 (20 µg/tumor/mouse) into the irradiated tumors, within 3 h after the irradiation. The control mice were injected with 50 µL of phosphate buffered saline (PBS) and treated mice with the same volume of anti-CD40. The irradiation and injection were done on anesthetized mice.

### 4.4. Immunoadjuvant Dosing Titration Study

We created six groups (5 mice/cohort) from mice with similar tumor sizes. There were control groups such as no treatment, single radiation dose of 6 Gy only, single administration of anti-CD40, and a group where anti-CD40 was intratumorally injected in 3 doses. The remaining two groups included treatment of a single dose of RT combined with anti-CD40 (total of 20 µg/tumor/mouse), administered either as single dose or three doses. The groups with a multiple dosing schedule were intratumorally injected on days 0, 7 and 14, after the treatment initiation. The administered volume of PBS (control and RT only) or anti-CD40 suspension was 50 µL. The mice cured from their initial tumors were re-challenged 100 days later with a subcutaneous injection of TC-1 cells (1 × 10^5^ cells), in comparison to new control mice.

### 4.5. Computed Tomography (CT) Imaging and Radiation Therapy (RT)

A Small Animal Radiation Research Platform (SARRP, Xtrahl, Inc., Suwanee, GA, USA) was used for the image-guided irradiation of mice in cohorts, including radiation at 220 kVp and 13 mA, using a 10 × 10 collimator and a 0.15 mm copper filter. The radiation doses varied according to the study design. In the radiation titration study, the doses applied were single doses of 6 Gy, 10 Gy or 15 Gy and in the remaining studies, 6 Gy. Prior to irradiation, mice were anesthetized and CT images were taken to precisely aim and treat tumors on the right flanks. Whole body CT images taken 2 weeks post treatment were analyzed by using Imalytics Preclinical Software [23] to measure the 3D volume of the tumors.

### 4.6. Antibodies

Monoclonal agonistic anti-CD40 (clone FGK, BioXell, Lebanon, NH, USA) was utilized as a main immune respond booster throughout the experiments. The suspension of mAb and PBS was administered either as a single dose of 20 µg/tumor or as a three 6.7 µg doses administered on days 0, 7 and 14 post treatment. This dosing schedule was applied in the immunoadjuvant dose titration study. In the remaining studies, anti-CD40 was administered as a single dose of 20 µg into the one tumor. Mice treated with both RT and anti-CD40 were administered with anti-CD40, within 3 h after irradiation.

### 4.7. Tumor Volume Assessment

Tumor measurements were taken prior to treatment and then afterwards at least twice a week using a digital Vernier caliper. The tumors were measured between the skin layers, since tumors were subcutaneous. The length (L) of the tumor was measured in parallel to the spine and the tumor width (W), at a perpendicular angle to the length dimension. The tumor volume was calculated as: tumor volume = [0.5 × L × (W^2^)]. The tumor volumes were plotted as a function of time after the treatment. Mice were monitored for any changes in tumor sizes (either growth or shrinkage). Mice were humanely euthanized when they reached study endpoints, tumor size 2 cm in diameter, or humane endpoints, like ulceration of the tumor or poor body condition, even before they reached the study endpoint.

### 4.8. Flow Cytometry

Tumors were harvested from mice and digested in PBS supplemented with 1.5 mg/mL of Collagenase A (Roche, Basel, Switzerland), 0.4 mg/mL DNase I (Roche), 5% FBS (Corning, Corning, NY, USA) and 10mM HEPES (Gibco, Dublin, Ireland), and incubated at 37 °C. Cells were dispersed into a single cell suspention utilizing a 70 micron cell strainer (Corning). For this study we used anti-mouse antibodies LIVE/DEAD Fixable Violet Dead Cell Stain Kit (Invitrogen, Carlsbad, CA, USA), anti-CD45-FITC [clone 30-F11] (BioLegend, San Diego, CA, USA), anti-CD8a-PE/Cy7 [53-6.7] (BioLegend), anti-CD4- PerCP/Cyanine5.5 [GK1.5] (BioLegend), anti-XCR1-PE [ZET] (BioLegend), and anti-CD335-AF647 [29A1.4] (BioLegend), at determined titration concentrations. Samples were analyzed using a BD LSR Fortessa cytometer, and collected data were analyzed with FCS Express 6 software (De Novo Software, Pasadena, CA, USA).

### 4.9. Statistical Analysis

Tumor volumes were statistically analyzed by using the standard Student’s two-tailed t-test. All results with *p*-values * *p* < 0.05, ** *p* < 0.01, *** *p* < 0.001 were considered as statistically significant. Mice survival data were plotted in GraphPad Prism 7.0 (GraphPad Software, San Diego, CA, USA) and analyzed for their significance utilizing a log-rank (Mantel–Cox) test. Error bars are SEM.

## 5. Conclusions

We demonstrate preclinical data of the abscopal effect in metastatic cervical cancer in the animal model. This rare abscopal effect was consistently observed for a single dose optimized at 6 Gy radiotherapy, in combination with locally delivered anti-CD40 inhibiting tumor growth, or complete remission in all tumor lesions. Further investigations to optimize and advance the mechanistic understanding of the abscopal effect associated with cervical cancer could lead to innovative treatment methods, offering a new more accessible treatment option for cervical cancer care. This could be particularly beneficial in many LMIC where cervical cancer remains the leading cause of cancer death in women and in addressing disparities in access to treatment in resource-poor settings.

## Figures and Tables

**Figure 1 cancers-12-01026-f001:**
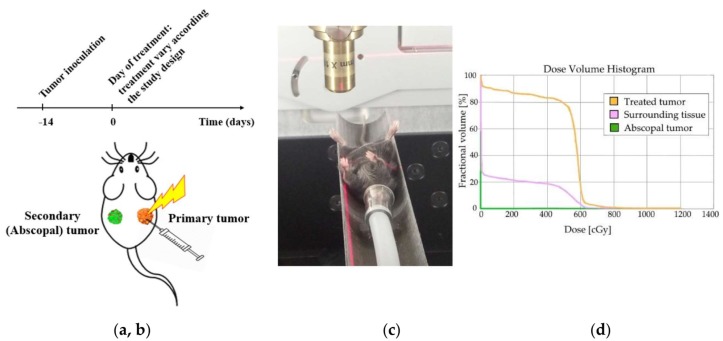
Study Design. (**a**) General timeline of the studies, including tumor inoculation 2 weeks prior to a treatment varying according to the study designs. In all studies, the treatment response was monitored over time by tumor volume measurements. (**b**) Schematic depiction of the treated mouse. Orange (right) tumor represents the treated side and green (left) tumor, the abscopal side. (**c**) Small Animal Radiation Research Platform set-up was used for RT and CT imaging. Image guided RT was only given to the right tumor. (**d**) Dose volume histogram showing RT dose distribution into the treated tumors and surrounding normal tissue, including contralateral non-treated tumors.

**Figure 2 cancers-12-01026-f002:**
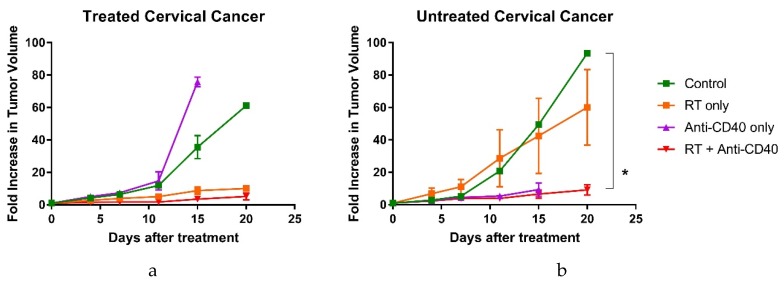
Investigating the abscopal effect in cervical cancer mouse model represented by tumor volume (mean ± SEM) in (**a**) treated and (**b**) untreated groups (*n* = 5). The abscopal effect was observed in combination treatment group with a significant tumor growth control in untreated and treated tumors. * *p* < 0.05.

**Figure 3 cancers-12-01026-f003:**
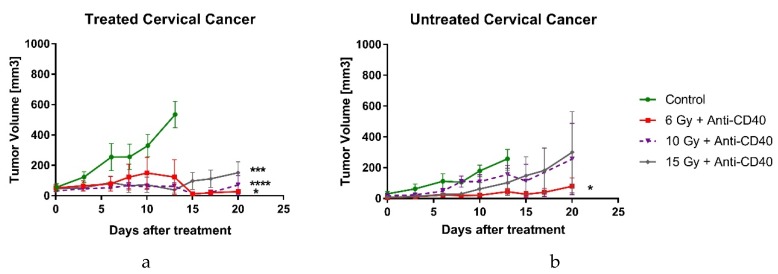
Radiation dose titration study in cervical cancer. C57BL/6 mice were subcutaneously inoculated with TC-1 cells on both flanks. Once tumors reached a 3–4 mm size, the right tumors were treated with a single dose of 6 Gy, 10 Gy or 15 Gy, with a combination of agonistic anti-CD40 (20 µg/tumor) (**a**) Volumes of treated tumors over time. Higher doses of RT had a better treatment impact, but did not induce the abscopal effect. (**b**) Volumes of untreated (abscopal) tumors over time. Single dose of 6 Gy combined with anti-CD40 induced abscopal effect on contralateral tumor, by inhibiting tumor progression. Error bars are SEM. *n* = 5 mice/group. * *p* < 0.05, *** *p* < 0.001, **** *p* < 0.0001.

**Figure 4 cancers-12-01026-f004:**
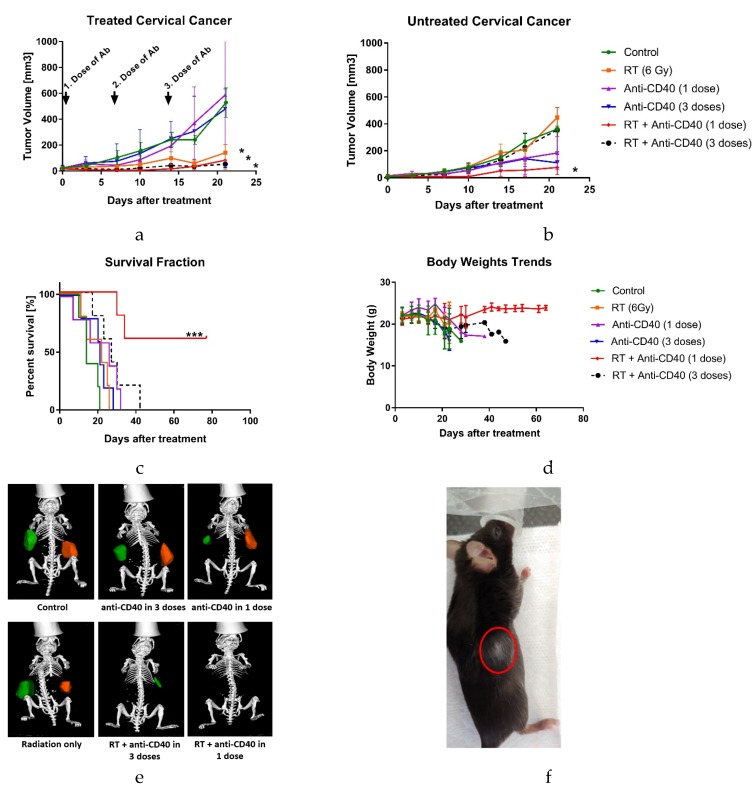
Immunoadjuvant dosing titration study on TC-1 tumors. Single dose of 6 Gy RT combined with anti-CD40 in single direct injection (concentration 20 µg/tumor) induced an abscopal effect, autoimmune memory, and autoimmune vitiligo. Other dosing regimens in combination with RT only effected the treated tumors. Once tumors reached treatable size, the right tumors were treated with single dose of 6 Gy or with a combination of agonistic anti-CD40, either in one dose or in three fractionated doses (day 0, 7, 14 post treatment) with a total application of 20 µg mAb. (**a**) Volumes of treated tumors over time. * *p* < 0.05; (**b**) Volumes of untreated (abscopal) tumors over time. Error bars are SEM. * *p* < 0.05; (**c**) Overall survival. *n* = 5 mice/group. *** *p* < 0.001 (**d**) Body weight averages of all groups. Increasing body weight corresponds to better mouse condition and smaller/no tumors. (**e**) Representative CT images of mice, two weeks after treatment. Orange color correlates to treated tumor and green to non-treated tumor. (**f**) Representative picture of vitiligo that occurred in mice in combination group of RT and single direct injection of anti-CD40 (20 µg) that had been re-challenged with TC-1 tumors.

**Figure 5 cancers-12-01026-f005:**
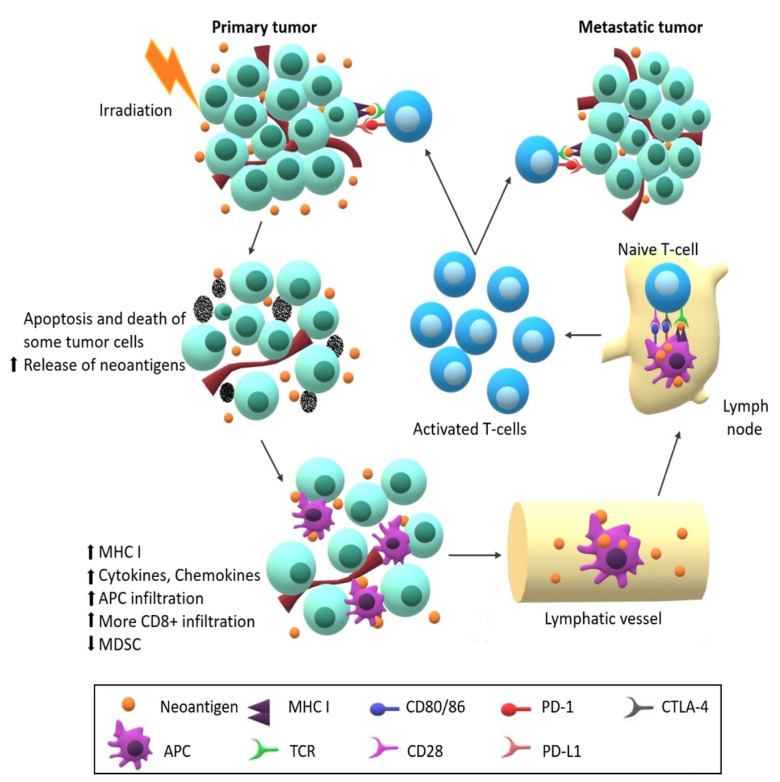
Scheme of Abscopal Effect. Irradiation of the cancer cells leads to apoptosis and immunogenic cell death, which contributes to higher release of tumor neoantigens. Radiation causes decrease of myeloid-derived suppressor cells (MDSCs), increased expression of major histocompatibility complex I (MHC I) on tumor cells and production of cytokines and chemokines attracting T-cells. Released neoantigens are up-taken by APCs, migrate through lymphatic vessels to lymphatic nodules, where they are presented to naïve T-cells by interaction of MHC I with T cell receptor (TCR). Naïve T-cells become activated upon further co-stimulatory signals, such as CD80, CD40 and CD28. Activated T-cells, mainly effector T-cells, proliferate and differentiate into neoantigen-specific cytotoxic T cell. These cells then migrate from lymph nodes and accumulate in the places with the neoantigen “label”, meaning all primary and metastatic tumor lesions. The abscopal effect could be hampered by presence of cytotoxic T-lymphocyte associated antigen 4 (CTLA-4) or programmed-death ligand 1 (PD-L1), which have immunosuppressive effect.

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
