# Peer review of "Single Radiotherapy Fraction with Local Anti-CD40 Therapy Generates Effective Abscopal Responses in Mouse Models of Cervical Cancer"

_cancers, 2020, doi:10.3390/cancers12041026_

Round 1

Reviewer 1 Report

The current research tested the hypothesis that the combination of radiotherapy with in-situ administration of the immunoadjuvant anti-CD40 can induce abscopal response in mouse models of cervical cancer. The study investigated clinically significant questions, but major revision is required to draw definite conclusions.

  1. Although the agonistic effect anti-CD40 may not be relevant to the expression of CD40 on cancer cells, would different expression levels of CD40 of cancer cells influence its synergistic or abscopal effect with radiotherapy?
  2. In order to conclude the tumor control effect of abscopal tumor is induced by immune response, please do further experimental analysis, such as immnohistochemistry or immune profiling for the primary and secondary tumor to investigate its immune response.

Reviewer 2 Report

The authors attempt to show the efficacy of radiotherapy with local anti-CD40 therapy in a cervical cancer xenograft model. This study is interesting; however, there are several critical problems to resolve. The reviewer’s comments are listed below.

The authors should not use a t-test for the statistical analysis of tumor volumes.

I would recommend that the authors use multiple cell lines to enhance the robustness of their results.

Did the TC-1 cells show the expression of CD40?

The authors suggest a hypothesis related to the abscopal effect of CD40. Because Cancers is a high-impact journal, the authors should attempt to investigate and explain the mechanism at a much deeper level.

Figure 2

What does the asterisk indicate?

In treated cervical cancer mice, the tumor volume in the anti-CD40 mice seemed to be significantly larger than that in the control mice. Please provide an explanation for this finding.

Figure 3

The authors measured the tumor volume of control group mice until only 13 days after treatment. They should continue the observation until 20 days after treatment.

Discussion

The authors should investigate the mechanism of the abscopal effect of CD40. These investigations would represent the key results in their study. The readers will not be satisfied with only a scheme of a hypothesis.

Conclusion

The authors focus on the treatment of cervical cancer in low-middle income countries (LMICs). I think this is an important assessment. As the authors pointed out, cervical cancer remains the leading cause of cancer death in women in LMICs. However, in my experience, irradiation is not often performed in these countries due to the lack of appropriate medical facilities and even pathological examination for cervical cancer is difficult in LMICs. Of course, antibody treatment is also quite difficult. What are the authors’ thoughts on this problem?

Author Response

Dear Reviewer,

Thank you, 

Jana Wood

Round 2

Reviewer 1 Report

The authors have provided responses adequately.

Reviewer 2 Report

Dear authors,

Thank you for your effort to revise the manuscript. The reviewer thinks the authors could revise the manuscript well.

Best,